# The Roles of the Colon Cancer Associated Transcript 2 (CCAT2) Long Non-Coding RNA in Cancer: A Comprehensive Characterization of the Tumorigenic and Molecular Functions

**DOI:** 10.3390/ijms222212491

**Published:** 2021-11-19

**Authors:** Radu Pirlog, Rares Drula, Andreea Nutu, George Adrian Calin, Ioana Berindan-Neagoe

**Affiliations:** 1Research Center for Functional Genomics Biomedicine and Translational Medicine, “Iuliu Hatieganu” University of Medicine and Pharmacy, 400012 Cluj-Napoca, Romania; pirlog.radu@yahoo.com (R.P.); drula.rares@gmail.com (R.D.); andreeanutu.an@gmail.com (A.N.); 2Department of Translational Molecular Pathology, Division of Pathology, The University of Texas MD Anderson Cancer Center, Houston, TX 77030, USA; gcalin@mdanderson.org

**Keywords:** CCAT2, lncRNA, competitive-endogenous RNA, miRNA, non-coding RNA, cancer

## Abstract

Colon cancer-associated transcript 2 (CCAT2) is an intensively studied lncRNA with important regulatory roles in cancer. As such, cumulative studies indicate that CCAT2 displays a high functional versatility due to its direct interaction with multiple RNA binding proteins, transcription factors, and other species of non-coding RNA, especially microRNA. The definitory mechanisms of CCAT2 are its role as a regulator of the TCF7L2 transcription factor, enhancer of MYC expression, and activator of the WNT/β-catenin pathway, as well as a role in promoting and maintaining chromosome instability through the BOP1–AURKB pathway. Additionally, we highlight how the encompassing rs6983267 SNP has been shown to confer CCAT2 with allele-specific functional and structural particularities, such as the allelic-specific reprogramming of glutamine metabolism. Additionally, we emphasize CCAT2’s role as a competitive endogenous RNA (ceRNA) for multiple tumor suppressor miRNAs, such as miR-4496, miR-493, miR-424, miR-216b, miR-23b, miR-34a, miR-145, miR-200b, and miR-143 and the pro-tumorigenic role of the altered regulatory axis. Additionally, due to its upregulation in tumor tissues, wide distribution across cancer types, and presence in serum samples, we outline CCAT2’s potential as a biomarker and disease indicator and its implications for the development of resistance against current cancer therapy regiments and metastasis.

## 1. Introduction

While the existence of non-coding species of RNA has been acknowledged for several decades already, the advent of modern sequencing techniques has allowed their detailed characterization and classification. As such, ncRNAs have been mainly categorized based on size, structure, and specific cellular function [1]. Smaller species of ncRNA are mainly represented by microRNA (miRNA), which are considered to range between 19 and 25 nucleotides and are mainly involved in regulating gene expression post-transcriptionally, targeting specific mRNAs based on the presence of complementary seed sequences [2,3,4,5]. On the other hand, long non-coding RNAs are a more recent class of non-coding RNAs, longer than 200 nucleotides, that are involved in a wide array of biological processes, ranging from early phase physiological developmental to intricate disease-associated and molecular cancer mechanisms [6,7]. Unlike miRNAs, lncRNAs are subjected to polyadenylation and splicing, and they mostly display preferential nuclear localization and highly cell-specific expression patterns. As such, lncRNAs have been widely investigated in different developmental [8,9] and pathological processes [10,11], which indicated their high functional variability. As a relevant example, the lncRNA HOTAIR, initially highlighted as a trans epigenetic regulator [12] of the key developmental *HOX* genes through its direct interaction with the *Polycomb Repressive Complex 2 (PRC2)*, was proven several years later to be a bona fide tumor-promoting lncRNA through the alteration of its regulatory mechanism in the case of metastatic breast cancer. In addition, the lncRNA NEAT2 (also known as MALAT-1) was initially indicated as a prognostic indicator for early-stage non-small-cell lung cancer [13]. Later investigations confirmed its regulatory involvement in mRNA metabolism based on the association with the SC35 [14] and serine/arginine (SR) splicing factors [15].

Colon cancer-associated transcripts 1 and 2 (CCAT1 and CCAT2) are two of the more recently identified transcripts with reoccurring implications in different types of cancers. First described in colon cancer, CCAT1 and CCAT2 both originate from the 8q24.21 chromosomal region, in close proximity with the *MYC* gene (500 kb and 300 kb upstream, respectively) and encompassing the cancer risk-associated rs6983267 single nucleotide polymorphism (SNP) (Figure 1) [16,17,18].

CCAT1 was the first one to be identified of the two, highlighted as a potential pre-malignant indicator for colorectal cancer (CRC), being identified as overexpressed in pre-tumoral tissues (adenomatous polyps and tumor-proximal colonic epithelium) but also in later stage tumor tissue [18]. Further investigations indicated that the *CCAT1* gene encodes two isoforms for the lncRNA, a long one (CCAT1-L), with preferential nuclear localization, and a short (CCAT1-S) isoform, mainly localized in the cytoplasm, which is presumed to be a processing result of the long variant [17,18,19]. Functionally, CCAT1 is a transcriptional regulator for *MYC* and acts as a long-distance enhancer element via its interaction with the transcription factor and chromatin organizer CTCF, thus promoting and stabilizing chromatin looping formation at the *MYC* locus [20]. CCAT1 is elevated in various types of cancers, including gastrointestinal cancers, hepatocellular carcinoma, lung cancer, melanoma, and multiple myeloma, where it was correlated with the activation of essential cancer biologic processes including proliferation, invasion, metastasis, and treatment resistance [18,19,21]. Other identified cancer-specific mechanisms involve the formation of a complex with the TP63 and SOX2 with transcription factors. The triplex can then bind both the CCAT1 promoter, determining its transcription, and the super-enhancer domains of epidermal growth factor receptor (EGFR), thus activating downstream MAPK signaling cascade with the effect of promoting small cell carcinoma tumorigenesis [22,23]. CCAT1 shows many functional similarities with its sister transcript, as we highlight further throughout this review.

Similarly, CCAT2 was first identified in colon cancer during investigations aiming to identify the functional elements associated with the rs6983267 SNP [16], a locus on the 8q24 “gene desert” often associated with increased predisposition to the development of colon, ovarian, and prostate cancers [24]. Referring specifically to the encompassing rs6983267 SNP, the genetic variants identified in multiple populations consist of the three GG, TT, and GT alleles, the latter being considered the wild type. The presence of a specific allele appears to have functional effects, which are discussed further in the latter chapters of this manuscript. Since then, CCAT2 has been investigated in multiple oncologic malignancies and is emerging as a relevant lncRNA. Therefore, the implications of CCAT2 in different cancer types, emphasizing all the influential cancer-specific mechanisms and highlighting its potential biomarker and therapeutic use, are within the scope of this review.

## 2. Cancer-Associated Regulatory Activity of CCAT2

### 2.1. MYC and WNT Pathway Activation

The first indication of CCAT2 being a tumor-associated transcript came from the study of its encompassed SNP, rs6983267. Ling et al. highlighted that the novel transcript, originating from the 8q24.21 chromosomal region, was significantly enriched in microsatellite-stable colon cancer samples. Initial in vitro and in vivo investigations confirmed that overexpressing CCAT2 increased both the proliferation and metastatic potential of colon cancer cells. Based on its genomic proximity to MYC, the group investigated whether CCAT2 had any regulatory effect on its expression [16]. Multiple validation studies confirmed a positive correlation between CCAT2 and MYC, both at the RNA and protein level, also indicative of a similar, although unconfirmed, enhancer activity similar to CCAT1 and a previous indication of chromatin looping associated with the rs6983267 locus [25]. Further investigations of the regulatory mechanisms indicated that the TCF7L2 transcription factor interacts directly with CCAT2, an event confirmed by immunoprecipitation and supported by its nuclear co-localization. Previous studies highlighted the regulatory activity of TCF7L2 on the expression of MYC via the presence of multiple TCF7L2 binding sites on the MYC promoter. Moreover, TCF7L2 is a known activator of the pro-tumorigenic Wnt/β-catenin, one of the ubiquitous pathways constitutively overactivated and considered a critical driver of CRC. On the other hand, MYC is one of the downstream targets of the Wnt/β-catenin pathway, highlighting the dual potentiating role of CCAT2 in this tumor-promoting feedback loop. Moreover, downstream of the Wnt/β-catenin pathway, the Wnt-induced-secreted-protein-1 (WISP1) was also significantly upregulated in CCAT2 active cancers. In the downstream pathway, high WISP1 levels stimulate the expression of its effectors VEGF-A, MMP2, and ICAM-1, with important roles in angiogenesis, epithelial-to-mesenchymal transition (EMT) and metastasis [26]. This interplay between CCAT2, MYC, and WNT molecules has been explored and confirmed in multiple cancers in recent years and is responsible for the main pro-tumorigenic roles of CCAT2. Similar studies confirmed the presence of the mechanism in the case of thyroid [27], ovarian [28], clear cell renal cell carcinoma [29], and breast cancers [30].

### 2.2. Allele-Specific Metabolic Reprogramming

As previously mentioned, the allelic variant defined by the rs6983267 SNP has functional effects on the secondary structure and the enrichment levels of CCAT2. The first functional differences were first reported by Ling et al. in 2013 [16]. Following the characterization of the allelic distribution of rs6983267 in the studied cohorts, the group wanted to investigate whether the rs6983267 status affected CCAT2 expression in patient samples. While there were no noticeable differences in patient samples, cell lines with heterogeneous rs6983267 genotype expressed the GG variant at a significantly higher level. Furthermore, the group indicated that the variation of G and T alleles influenced the secondary structure of the lncRNA [16].

A more intricate characterization of the functional consequences of the allele distribution came several years later when Redis et al. indicated that CCAT2 promotes an allelic-specific metabolic reprogramming of glutamine metabolism in CRC [31]. Specifically, the interaction between the G-allele CCAT2 and two subunits (CFIm25 and CFIm68) of the cleavage factor I (CFIm) promoted the alternative splicing of the glutaminase (GLS) enzyme [31]. The upregulation of the glutaminase enzyme acted as a metabolic switch resulting in an acceleration in glutamine metabolism, thus promoting proliferative and metastatic programs, confirmed both in vitro and in vivo. Additionally, the CCAT2-G-CFIm-GLS regulatory axis was identified in more than half (61%) of the investigated clinical samples, highlighting the potential clinical utility of this allelic-dependent process.

### 2.3. Chromosomal Instability

The first indication of CCAT2s as a promoter of chromosomal instability was highlighted in 2013 when Ling. et al. reported that CRC cell lines that overexpressed CCAT2 displayed a higher percentage of abnormal metaphases and aberrant centrosomes numbers. Additionally, the group reported a positive correlation score between CIN score and CCAT2 expression levels in 218 breast cancer patients with lymph node-negative disease [16]. Further details on the related mechanism were reported several years later in the case of microsatellite stable CRC, with the extending implications of the development and sustainability of cancer-associated chromosomal instability in the therapeutic resistance to 5-fluorouracil and oxaliplatin. Investigations confirmed that the mechanism is dependent on CCAT2’s interaction with the BOP1 ribosomal biogenesis factor, which were both upregulated as a result of CCAT2’s upregulation of MYC and stabilized following a direct interaction. Upregulation of BOP1 increased the active form of Aurora kinase B, causing chromosomal mis-segregation errors. Inhibiting this regulatory axis, both CCAT2 or BOP1 knockdown decreased the invasive phenotype of the cells both in vitro and in vivo. Additionally, higher expression of CCAT2 and BOP1 correlated with shorter survival times in CRC patients [32].

### 2.4. RNA Editing and CCAT2-Associated RNA–DNA Differences at the rs6983267 Locus

The concept of RNA editing consists of a conserved process of post-transcriptional modifications in the RNA transcripts that result in varying degrees of sequence modifications when compared with the originating DNA sequence [33]. Both coding and non-coding RNAs are subjected to different types of RNA editing, resulting in increased transcriptional diversity and various functional effects, dependent on the type of RNA modified and the nature of the modification. The general pattern (also known as canonical editing) of modifications consists of adenine to inosine (A to I) and cytosine to uracil (C to U) and is dictated by adenosine (ADAR) and cytidine (APOBEC) deaminases, respectively. Adenine to inosine is the most commonly occurring ADAR adenosine deaminase, mainly represented by ADAR1 and ADARB1, catalyzing the RNA editing predominantly in RNA duplexes formed in either the UTR regions of mRNAs, introns, and repetitive sequences, such as Alu repeats [34].

CCAT2 is subjected to a particular non-ADAR, non-APOBEC type of editing, specifically at the rs6983267 locus. Shah et al. first indicated the existence of a DNA-to-RNA allelic imbalance after observing discrepancies between the genomic DNA (gDNA) and the reverse-transcribed CCAT2 RNA sequences (cDNA) originating from the same myelodysplastic syndrome patients [35]. After the exclusion of any sequencing or technical errors, the group confirmed that the event, named RE (RNA Editing), was occurring specifically at the rs6983267 locus and consisted in the variation between the G and T alleles of CCAT2 not falling into the previously described canonical RNA editing patterns. Additionally, other non-physiological allele variations, such as C or A, were not identified. Further investigation indicated that the editing event was disease-specific, being identified at a much higher rate in the blood marrow and peripheral blood of myelodysplastic syndrome (MDS) and myeloproliferative syndrome (MPS) patients when compared with healthy controls. Further in vivo investigations confirmed the similar occurrence rate of RE in CCAT2 transgenic models, which was also correlated with increased splenomegaly and blood marrow hypercellularity in individuals in which RE was identified (RE+). As for the in vivo functional implications of rs6983267-RE, bone marrow-derived cells from RE+ mice were found to have significant alterations in immune-related pathways, especially genes involved in B-cell receptor, IL22, and IL4 signaling. Further immune profiling of RE+ hematopoietic stem and progenitor cells highlighted D22, CD19, and H2-Ob as the main dysregulated genes in both cell types. All things considered, while the exact RNA editing mechanism to which CCAT2 is subjected has not yet been uncovered, there is significant evidence pointing towards its role in the modulation of immune signaling in MDS and MPS [35].

### 2.5. ceRNA Activity of CCAT2 and miRNA Sponging

A commonly described role of lncRNAs in different pathologies is their complementarity-based interaction with different species of RNA, especially miRNAs [36,37], thus creating complex regulatory networks that involve both the interacting miRNAs and their mRNA targets [38]. Also called “miRNA sponging”, the competitive endogenous RNA mechanism (ceRNA) implies the competitive binding via a complementary sequence of a single or multiple miRNAs to the lncRNA rather than its mRNA target, thus mitigating the regulatory function of that miRNA [39]. The miRNA binding sites are named MRE (miRNA response elements) and consist of short sequences (under 10 bp) in the lncRNA transcript [40].

Recent studies have also highlighted CCAT2 as having multiple miRNA regulatory targets (Figure 2, Table 1), acting as one of the main pro-neoplastic drivers in various human cancers [39,41,42,43]. For example, CCAT2 promotes its neoplastic regulatory functions through extensive ceRNA networks involving multiple tumor suppressor miRNAs, including miR-424, miR-145, miR-23b-5p, and miR-143 [39,42,44,45,46].

In glial tumor cells, CCAT2’s role in angiogenesis and apoptosis was investigated through mechanistic studies that revealed that CCAT2 is encapsulated by tumor cells in exosomes, which are subsequently internalized by endothelial cells. Exosomes enriched in CCAT2 promoted the translation of proangiogenic genes (*VEGF*, *TGF β*, *FGF*, and *KDR)* and antiapoptotic gene *BCL-2* by inhibiting the proapoptotic genes *BAX* and *CASPASE-3*. Following the CCAT2 activation, the endothelial cells enhance the angiogenesis pathway, which leads to downstream upregulation of VEGF, TGFβ, and FGF protein secretion. Moreover, Lang et al. showed that the inhibitory effect on apoptosis of endothelial cells is enhanced under hypoxic conditions. Endothelial cells pre-exposed to hypoxia and then transfected with CCAT2-enriched exosomes showed increased expression of *BCL-2* and lower *BAX* and *CASPASE-3* expression [71]. Additional studies revealed that part of the CCAT2 regulatory effect on the *VEGF* gene is modulated through direct induction of the PI3K/AKT signaling pathway and via sponging of the miR-424 [43]. The ceRNA regulatory network is based on both CCAT2 and the VEGFA mRNA sharing a miR-424 MRE [43]. This ceRNA relationship between CCAT2 and miR-424 was first described in epithelial ovarian cancer, where the tumorigenic effects of CCAT2 were found to be modulated via the inhibition of the tumor suppressor activity of the miR-424. Knockdown of CCAT2 reduced cellular proliferation, migration, and angiogenesis through upregulation of miR-424 and consequent inhibition of VEGF, confirming that CCAT2 is a crucial element of the processes via this regulatory axis [39].

A new regulatory axis between CCAT2/miR-424 and the *protein checkpoint kinase 1 (CHK1)* gene was described in glioma, with roles in the tumor chemoresistance [44]. Previous studies reported that miR-424 tumor suppressor activity relies on interfering with genes (including *CHK1*) that are essential to the G1/S cell cycle transition, thus limiting cellular proliferation and promoting apoptosis. In cervical cancer, miR-424 expression was inversely correlated with the *CHK1* gene, and high Chk1 protein levels and reduced levels of miR-424 were associated with aggressive behavior and poor outcome [101].

EMT is responsible for tumor aggressive behavior and acquisition of the ability to develop distant metastasis. CCAT2 is an enhancer of EMT and cellular proliferation through downregulation of E-cadherin and upregulation of N-cadherin and vimentin [28,62]. Additionally, CCAT2 was found to modulate the EMT phenotype through cAMP-responsive element-binding protein 1 (*CREB1*) expression modulation [96]. *CREB1* is an oncogene that was shown to promote tumor cell proliferation, EMT transition, and metastasis in gastric cancer, CRC, and prostate cancer [102,103]. CREB1 was shown to be a target of the tumor suppressor miR-493-5p. In prostate cancer cell lines, upregulation of miR-493-5p inhibited *CREB1* expression and inhibited EMT via AKT/GSK-3β/Snail signaling [103]. Wang et al. investigated the regulatory loop between CCAT2/miR-493-5p/CREB1 in cervical cancer, revealing that CCAT2 sponges miR-493-5p leading to upregulation of *CREB1*, which promotes proliferation and EMT [96].

Bioinformatic sequence prediction analysis revealed the possible interaction between CCAT2 and miR-23b-5p based on a putative binding site in the 3′-untranslated region (3-UTR) of CCAT2. miR-23b-5p was described as acting as a tumor suppressor in a variety of cancers, most importantly by inhibiting the *FOXC1* gene, a known oncogene with important roles in signal transduction and tumor progression [104,105]. The direct interaction between CCAT2 and miR-23b-5p was evidenced using the dual-luciferase reporter assay in non-small-cell lung cancer cell lines (NSCLC). The expression level of CCAT2 was further investigated in relation to the expression of miR-23b-5p target gene *FOXC1*. A direct correlation between CCAT2 expression level *FOXC1* gene expression and an inverse correlation between CCAT2 and miR-23b-5p was described. These data support that CCAT2 coordinates a loop that enhances the expression of the *FOXC1* gene by direct sponging of the tumor suppressor miR-23b-5p. Moreover, the increased levels of FOXC1 further promote cellular proliferation and migration [42].

In hepatocellular carcinoma (HCC) cell models, CCAT2 was reported to promote cellular migration, proliferation, and decreased apoptosis by interaction with another member of the *FOX* gene family, *FOXM1* [65,66,106]. Referring to *FOXM1,* it was previously shown to be an important tumor promoter in HCC progression, and its upregulation is correlated with a poor prognosis [106]. Recent investigations suggest a connection between CCAT2 and FOXM1 via a regulatory loop involving miR-34a. The study highlighted that increased levels of *FOXM1* were correlated with CCAT2 in HCC cell lines. Bioinformatic analysis indicated miR-34a as the regulatory link between CCAT2 and FOXM1, as both transcripts have the competing binding ability for miR-34a. The direct interaction between CCAT2 and miR-34a was evidenced by dual-luciferase reporter assay analysis using miR-34a and CCAT2-WT and CCAT2 miR-34a mutated binding site [66]. Thus, overexpression of CCAT2 might be responsible for the mitigation of the tumor suppression ability of miR-34a, which has been also shown to promote the activation of cellular senescence programs by downregulating the *FOXM1* oncogene independent of CCAT2 [107].

A different study on the role of CCAT2 in HCC has highlighted its involvement in autophagy and metastasis programs via a bimodal cytoplasmic and nuclear mechanism [69]. The CCAT2 localization in the HCC cell cytoplasm promotes its ceRNA interaction with miR-4496, a previously described tumor suppressor [69]. On the other hand, CCAT2 interacts with the RNA binding protein (RBP) ELAVL1/HuR at the nuclear level, a key factor involved in mRNA stabilization, previously shown to promote proliferation, apoptosis, and differentiation in multiple types of cancer [108]. Shi et al. showed that ELAVL1 knockdown decreased Agt5 expression, this effect being restored through CCAT2 upregulation. These findings indicate that CCAT2 promotes the proliferative processes by targeting miR-4496 and the activation of autophagy via, and interacting with ELAVL1, and downregulating Agt5. [69].

One of the CCAT2s main regulatory ceRNA targets is miR-145, a tumor suppressor described in various solid human cancers, including colon, gastric, and prostate cancers [109,110]. In these cancers, miR-145 inhibits tumor proliferation by inhibiting the *MYC* and *Friend leukemia virus integration 1 (FLI-1)* genes [111,112]. The opposite regulatory effects between miR-145 and CCAT2 on the *MYC* pathway suggested a possible modulatory relationship between these two ncRNAs that was further investigated. Considering miR-145 biogenesis and using bioinformatic analysis, interaction with the pre-miR-145 was found. This relationship was confirmed on cellular models that support downregulation of the mature miR-145 by direct interaction between CCAT2 and pre-miR-145 at the nuclear level [49]. Additionally, a direct bidirectional modulatory relationship between miR-145 and the oncogenic miR-21 was shown, in which miR-145 negatively regulates miR-21 in colon cancer cell lines [113]. Therefore, by inhibiting miR-145 maturation, CCAT2 indirectly upregulates the miR-21, which promotes proliferation and is associated with a poor prognosis [49].

The CCAT2/miR-145 regulation axis was also described in HCC progression via modulation of the *MDM2* gene [45]. *MDM2* is an oncogene previously reported in epithelial cancers, high-grade tumors, being a signature biomarker in multiple soft tissues [114]. In HCC, the MDM2–p53 pathway is an essential pathway in tumorigenesis, being altered in 25% of cases [115]. In physiological conditions, p53 enhances miR-145 activity, which was shown to act on MDM2. In human epithelial cancers, when an alteration of the p53 gene is produced, the miR-145 is downregulated, which removes the inhibition on the MDM2 oncogene [116]. In HCC, miR-145 expression was inversely correlated with MDM2 and was shown to directly interact with it by dual-luciferase reporter assay [45].

Further analysis into the mechanisms by which CCAT2 promotes human esophageal carcinoma cell (ESCC) progression led to the identification by bioinformatics analysis of miR-200b as a possible target. The direct interaction between CCAT2 and miR-200b was confirmed by a dual-luciferase reporter assay [60]. Functionally, miR-200b is a tumor suppressor miRNA that was reported in ESCC to act as an inhibitor of cellular proliferation and invasion [117,118]. Moreover, miR-200b was shown to have a complementarity sequence with the insulin-like growth factor 2 mRNA-binding protein 2 (IGF2BP2) oncogene [60]. IGF2BP2 and thymidine kinase 1 (TK1) are two important oncogenes with aberrant expression in ESCC that are associated with advanced disease and poor prognosis. IGF2BP2 was shown to act as a reader on the N6-methyladenosine (m6a) to modulate tumor progression [119]. Wu et al. confirmed by in vitro experiments the indirect positive regulatory effect of CCAT2 on the IGF2BP2 through sponging of miR-200b. Moreover, the regulatory loop includes the modulation of TK1 mRNA methylation by the IGF2BP2, which stabilizes the TK1 and enhances its expression. Therefore, the tumor promoter effects of CCAT2 in ESCC are directed by the CCAT2/miR-200b sponging, which limits the miR-200b levels and inhibitory effect on IGF2BP2. Increased IGF2BP2 protein levels recognize TK1 m6a modification and maintain its stability, inducing cellular proliferation and invasion [60].

CCAT2 was highly expressed in endometrial cancer tissue and cell lines; its expression is correlated with increased cellular proliferation and metastasis. The antiapoptotic effect of CCAT2 is switched on by sponging off the tumor suppressor miR-216b. At normal levels, miR-216b is an inhibitor of the anti-apoptotic gene *BCL-2*. CCAT2 contains the binding site for miR-216b and was revealed to directly interact with the miRNA. The increased levels in CCAT2 via the inhibition of miR-216b enhanced the activity of BCL-2, which in turn activated the PTEN/PI3K/AKT and mTOR signaling pathways that promote cellular proliferation and malignant transformation [41].

As another ceRNA target of CCAT2, miR-143 is a tumor-suppressor miRNA with important roles in the inhibition of major oncogenes involved in proliferation and treatment resistance. miR-143 was investigated in colon, breast, and cervical cancers [46]. Bi et al. predicted by bioinformatic analysis that miR-143 was a possible target for CCAT2 [83]. The direct interaction between CCAT2 and miR-14 was revealed in osteosarcoma cell lines. Furthermore, miR-143 is known to interact with the *FOSL2* [120]. FOSL2 is a known oncogene that promotes metastasis and angiogenesis in breast and lung cancer [121,122]. Therefore, a regulatory axis between *FOSL2*, CCAT2, and miR-143 was described, with CCAT2 upregulation modulating metastasis and proliferation through indirect upregulation of *FOSL2* oncogene via sponging of the miR-143 [83].

### 2.6. Therapeutic Resistance

The expression level of CCAT2 was associated in several cases with chemotherapeutic and radiotherapeutic resistance in gliomas, CRC, thyroid, breast, and esophageal cancers [32,59,89,123,124]. The roles of lncRNA CCAT2 on modulating therapeutic resistance in these cancers are presented in Table 2.

The regulatory roles of CCAT2 in cancer therapeutic resistance are related to its direct inhibition of tumor suppressor miRNAs and through its effect on the DNA damage response pathway and chromosomal instability [32,44,59]. The regulatory axis between CCAT2/miR-424/*CHK1* was proven in gliomas, showing a direct correlation between CCAT2 expression direct inhibition of miR-424 and upregulation of *CHK1*. Interestingly, *CHK1* was described as an important modulator of chemotherapy and radiotherapy resistance in acute myeloid leukemia and glial tumors and via its effect in the DNA damage response [124,126,127]. The CCAT2 upregulation in glioblastoma cell lines was associated with upregulation of the *CHK1* gene and increased resistance to teniposide, temozolomide, vincristine, and cisplatin [44].

The previously described regulatory axis between miR-145 and CCAT2 was further investigated in ESCC lines, focusing on the axis’s role in promoting radiotherapy resistance. miR-145/p53 pathway downregulation was shown to promote radioresistance by altering the normal response to DNA damage and inhibiting apoptosis [128]. Additionally, miR-145 was shown to inhibit the expression level of P70 ribosomal protein S6 kinase 1 (p70S6K1), an important effector on the mTOR pathway that is involved in radioresistance [129,130]. p70S6K1 upregulation induces downstream activation of the Akt/ERK/p70S6K1 signaling pathway, enhancing cellular viability and proliferation [59]. Moreover, the enhancing effect of CCAT2 on radioresistance is influenced through its negative regulatory roles on p53 and p21 proteins, which are essential for cell cycle arrest, DNA repair, and promotion of apoptosis [59]. Targeting CCAT2 in ESCC demands further investigation to better understand its implications on tumor progression and involvement in tumor radioresistance [58].

### 2.7. CCAT2 Enhancer Activity and Interactions with RBPs

CCAT2 was upregulated in pituitary adenomas in comparison with adjacent normal tissue and associated with an aggressive tumor phenotype. CCAT2 overexpression was partly explained by the increased E2F1 factor, which binds to the CCAT2 promoter region and facilitates CCAT2 transcription. Increased CCAT2 levels were shown to inhibit degradation of the Securin protein, which could explain the aggressive phenotype of the pituitary adenomas [131]. Additionally, in a different study, inhibiting CCAT2 reduced proliferation, invasion, and increased cellular apoptosis—effects correlated with reduced expression of TGF-β, MAD homolog 2, and α-SMA proteins. These results highlight the CCAT2 regulatory role on the TGF-β signaling pathway [92]. Moreover, CCAT2’s positive effect on cellular proliferation is induced by suppressing p15 protein, a tumor suppressor protein that arrests the cell cycle in G1 by interacting with the histone methyltransferase EZH2 [91].

A recent study by Zang et al. investigated the mechanisms by which CCAT2 promotes cellular proliferation and showed that it directly binds to YAP protein and inhibits its phosphorylation, which further promotes YAP nuclear translocation and activation of YAP’s downstream oncogenic targets CTGF, CYR62, and AMOTL2 [132].

Investigations into the CCAT2 modulatory role on HCC proliferation and metastasis revealed that its expression is closely correlated with the *NDRG1* gene. CCAT2 promotes the expression of the *NDRG1* gene by enhancing its promoter activity [68]. *NDRG1* is an important oncogene in HCC and is involved in promoting proliferation, metastasis, and poor prognosis [133]. Similar to other cancers, upregulation of CCAT2 enhances the EMT and is associated with a poor prognosis. HepG2, SMMC772, and MHCC97H cell lines were used to investigate the promoter role of CCAT2 on the EMT showing that its expression is positively correlated with Vimentin and Snail2 and negatively correlated with E-cadherin [67].

In vitro downregulation of CCAT2 in pancreatic cell lines PANC-1, SW1990, and PC-3 hindered proliferative and invasive capabilities of the cells, presumably due to regulatory interaction with the mitogen-activated protein kinase (MAPK) pathway [70], an important pathway involved in the translation of the signals extracellular messengers to intracellular responses [23].

In neuroblastoma, a malignant embryonal solid tumor that most frequently affects children, CCAT2 was found at increased levels in tumor tissue compared with adjacent normal tissue, and its expression was associated with increased cellular proliferation and lower three-year survival. Chen et el. revealed that part of the CCAT2 effect on cellular proliferation is due to inhibition of tumor suppressor protein p53 and enhancement of the anti-apoptotic protein Bcl-2 [75].

CCAT2 upregulation promotes EMT by upregulating *ZEB2* gene expression and decreasing the expression of the *E-cadherin* gene. CCAT2 modulates EMT by interacting epigenetically with *EZH2*, *H327me3*, and *LSD1*. Increased occupancy of these genes decreases their binding affinity across the promoters of E-cadherin and LATS2. These results were validated on in vivo models in which inhibition of CCAT2 was associated with increased expression of E-cadherin and LATS2 and slower tumor growth [62].

Moreover, CCAT2 expression in gastric cancer modulates the expression of the *POU5F1B*, a retrogene located adjacent to MYC that inhibits apoptosis and stimulates angiogenesis and metastasis [134,135]. CCAT2 silencing is increasing apoptosis and autophagy in vitro through a downregulation of PI3K and mTOR pathways, suggesting that CCAT2 has also a regulatory effect on the PI3K/mTOR signaling pathways [63,134].

Inhibition of CCAT2 in ovarian cancer cell lines reduced invasiveness and cellular progression and inhibited EMT by downregulating vimentin and N-cadherin and upregulating E-cadherin [28]. Interestingly, treatment with calcitriol (vitamin D) on ovarian cancer cell lines had the same inhibitory effect that was achieved by silencing CCAT2. Calcitriol stimulation highlighted an inhibitory effect on CCAT2’s regulatory activity by decreasing the affinity between the transcription factor TCF7L2 and the *MYC* promoter, hindering its expression [98] and therefore supporting a possible place for calcitriol as an inhibitor of CCAT2 in cancer.

## 3. CCAT2 as a Potential Biomarker

CCAT2 expression levels and their influence on cancer progression have been assessed in various cancer types and compared with survival data and prognostic factors, including proliferation, tumor size, migration, and metastasis (Table 1) [50,136]. Various studies analyzed the association between CCAT2 expression level in cancer, identifying a direct association between CCAT2 upregulation and disease progression, invasion, positive lymph nodes, and metastasis [17,137]. When comparing the tumor tissue with adjacent normal tissue in solid cancers, CCAT2 expression was upregulated in the tumor tissue, supporting its role as a potential biomarker [17,137]. A meta-analysis by Jing et al. indicated the correlations between CCAT2 expression level in tumor tissue and the clinical stage and outcome and identified a positive correlation between CCAT2 expression level and the risk of developing distant metastasis, positive lymph nodes, shorter progression-free survival, and poorer overall survival [138].

CCAT2 expression was reported as being elevated in the plasma of cancer patients with cervical cancer, gastric cancer, colon cancer, lung cancer, and multiple myeloma [64,80,86,95,139]. The serum expression levels of a 3 lncRNA panel, which includes CCAT2, were able to distinguish early-stage cervical cancer patients from normal controls, with an area under the curve of 0.894, 67.1% sensitivity, and 96.1% specificity [95]. Therefore, this model validates a simple non-invasive procedure that has the potential to use serum samples to better identify early-stage cervical cancer. In gastric cancer, CCAT2 was investigated as part of a 5 lncRNAs (TINCR, CCAT2, AOC4P, BANCR, and LINC00857) plasma panel. Their lncRNA-based index managed to outperform the CEA-based index (*p* < 0.001) when distinguishing gastric cancer patients from healthy patients. Additionally, the expression levels of the lncRNAs part of the index decreased post-surgically, suggesting their role in dynamic monitoring of the patients for relapse [64]. In multiple myeloma, CCAT2 was upregulated in both peripheral blood and bone marrow when comparing patients with healthy controls. The CCAT2 expression level was correlated with International Scoring System stages, kidney function, and light chain concentrations. Moreover, when CCAT2 expression level was associated with classical multiple myeloma biomarkers such as IgA, β2MG, and HGB, the area under the curve (AUC) was significantly improved up to 0.974 (95% CI 0.958~0.990; *p* < 0.001). In addition, integrating the CCAT2 expression with classical biomarkers improved the sensitivity and specificity of multiple myeloma diagnoses [86].

In CRC, CCAT2 was investigated as part of 10 lncRNAs panels including CCAT1, H19, HOTAIR, HULC, MALAT1, PCAT1, MEG3, PTENP1, and TUSC7 as potential biomarkers for early detection of CRC from stool samples. The diagnostic performance of the combined lncRNA model when distinguishing the CRC from non-malignant samples had an area under the ROC curve of 0.8554 in the training set and 0.8465 in the validation set [52]. Although more in-depth studies on the intricate regulatory functions of CCAT2 in cancer are needed, it has the potential to be used as a prognostic biomarker and as a possible therapeutic target in CRC [50,139,140]. On the other hand, when CCAT1, CCAT2, and *MYC* expression levels were assessed as risk factors for predicting early-stage CRC metastasis, the results were limited for two lncRNAs. Even though, the individual expression of CCAT2 was individually upregulated in CRC tissue when compared with the adjacent control, its expression did not improve a 19 gene classifier, ColoMet19, previously proposed by the same authors for early-stage CRC metastasis. Nevertheless, CCAT2 expression was higher in metastatic patients compared with metastatic negative patients. Even though these results support the differential expression of CCAT2 in CRC samples when comparing with normal tissue and in metastatic vs non-metastatic CRC, they show that when integrating the CCAT2 expression level into the global genomic alterations that occur in cancer, its role fails to significantly influence the previously described ColoMet19 predictor [54].

## 4. Conclusions

The role of CCAT2 in either initiating or promoting the oncological phenotype of multiple cancer types is still an expanding niche in the area of tumor-associated non-coding RNAs. Currently, the main highlighted and reoccurring regulatory interactions with tumor-promoting signaling pathways are Wnt/β-catenin and MYC. Additionally, novel insight regarding CCAT2’s implication in promoting cancer-associated genomic instability hints towards the multiple implications of these transcripts across the various types of cancers associated with chromosomal abnormalities. It is worth mentioning both the direct (ceRNA activity) and indirect (transcriptional regulation) interactions with other species of ncRNAs, especially miRNAs, which add a layer of complexity in the regulatory axis in which CCAT2 is involved. In most described studies, CCAT2 was highlighted as a transcript with both biomarker and prognostic potential, as its overexpression was correlated with poor prognosis, presence of metastases, and reduced progression-free and overall survival. Therefore, CCAT2 could be considered a valuable pan-cancer biomarker that is associated with a more aggressive disease course, one with potential as a therapeutic target of new treatments or for use in strategies for overcoming chemoresistance and radioresistance [136,137,138].

Although most studies offer similar results regarding CCAT2 expression in cancer tissue and adjacent control, with significant differences between the study samples, recent studies on Saudi and Iranian populations that investigated CCAT2 expression in CRC and lung cancer failed to reproduce the results. The divergence in results should be further investigated in similar populations or studied in a larger sample to understand whether the cause was due to study sample characteristics or due to differences in etiology or lncRNA signatures [51,79].

## Figures and Tables

**Figure 1 ijms-22-12491-f001:**
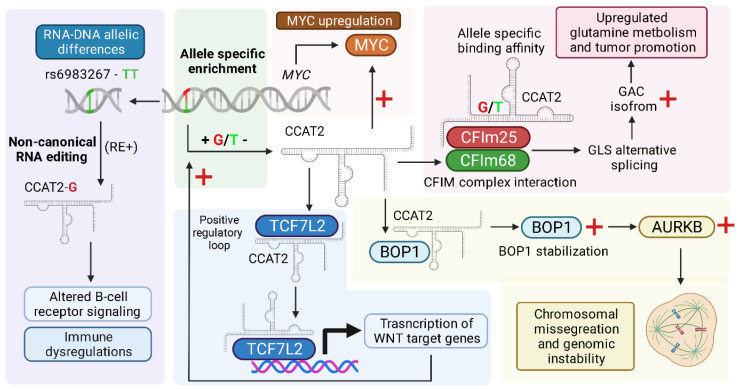
Overall schematic of the main tumor-promoting regulatory mechanisms of CCAT2 described to date.

**Figure 2 ijms-22-12491-f002:**
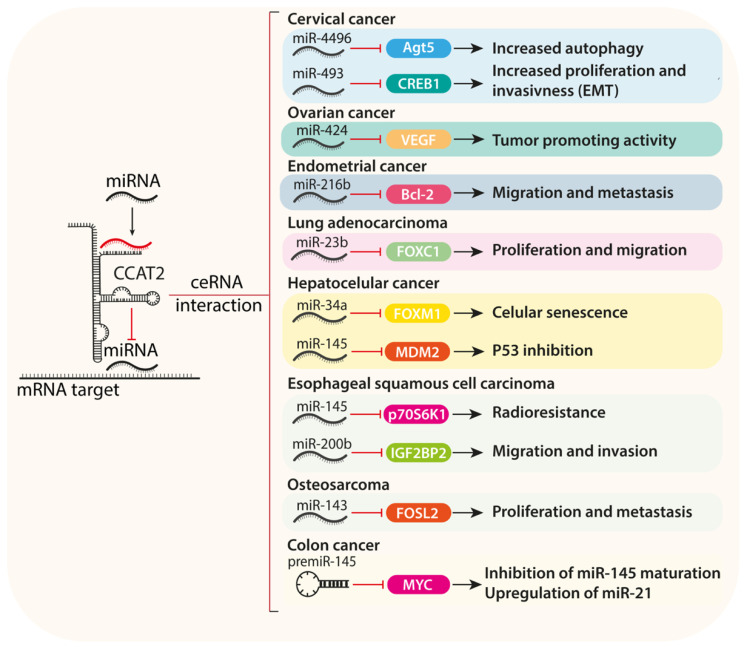
Currently known interactions of CCAT2 and other ncRNAs. The figure presents the effect of CCAT2 sponging on tumor suppressor miRNAs and how it influences essential cancer processes.

**Table 1 ijms-22-12491-t001:** Currently known regulatory targets and biological effects of different expression patterns of CCAT2 in various cancer types.

Cancer Type	Study Samples	CCAT2 Expression	Regulatory Targets	Biological Effect	Ref.
Colorectal cancer	Tissue: 191 CRC and ANTT Cell lines: COLO320D, MHCT116, RKO, HEK293	↑	↑Wnt/β-catenin ↑MYC, ↑*TCF7L2* ↑ MS status	Invasion, Distant Metastasis	[16]
	Cell lines: HCT116, KM12C, KM12SM, COLO320, DLD-1, HT29 In vivo model: CCAT2 transgenic mice, WT mice	↑	↑BOP1 ↑AURK B	Chromosomal instability, Chemoresistance to 5 fluorouracil and oxaliplatin	[32]
	Tissue: 218 CRC and ANTT	↑		Differentiation grade TNM stage, Lymph nodes metastasis, Distant metastasis, Vascular invasion, Poor prognosis	[47]
	Tissue: 149 CRC and ANTT	↑		Distant metastasis	[48]
	Cell lines: HCT-116, HT-29	↑	↓ pre-miR-145, ↑ miR-21	Proliferation, Invasion	[49]
	Tissue: 280 CRC and ANTT	↑	↑MS status ↑ MYC	Poor prognosis, Lymph nodes metastasis, TNM stage	[50]
	Blood: 63 CRC and 40 Controls	N/A	-	-	[51]
	Tissue: 60 CRC, 30 Colon polyps, and 60 non-cancers.	↑ along with CCAT1, CCAT2, H19, HOTAIR, HULC, MALAT1, PCAT1, MEG3, PTENP1, and TUSC7		Part of a stool lncRNA panel for CRC detection	[52]
	Tissue: 80 CRC and ANTT Cell lines: FHC, HT29, Lovo, HCT-116.	↑	-	Cellular growth, Proliferation, Antiapoptotic	[53]
	Tissue: 150 CRC and ANTT	↑	↑MYC	Metastasis	[54]
Esophageal cancer	Tissue: 229 ESCC and ANTT	↑	-	Poor prognosis, Lymph node metastasis, TNM stage	[55]
	Tissue: 57 ESCC and ANTT Cell lines: TE13, KYSE410, ECA109, TE1/N: HEEC	↑TE1, TE13, KYSE410 ↓ ECA109	-	CCAT2 expression correlated with smoking status	[56]
	Cell lines: Eca-109, EC9706, KYSE150, TE-1/N: HEEC	↑	↑BCL-2, ↓BAX, ↓CYCLIN D1, ↑Wnt pathway	Proliferation, Migration, Invasion	[57]
	Tissue: 62 OSCC and ANTT Cell lines: Tca8113, Cal27/hNOK	↑	↑CCND1, ↑MYC, ↑Wnt/β-catenin	Poor prognosis, Invasion proliferation, T stage differentiation	[58]
	Tissue: 60 ESCC and 21 esophageal mucosa Cell lines: HEEC, TE-1, TE-3, ECA109, KYSE410, KYSE520	↑	↓miR-145, ↑p70S6K1, ↓p53 pathway	Radiotherapy resistance, cellular proliferation.	[59]
	Tissue: 33 ESCC and ANTT Cell lines: KYSE-410, KYSE-150, TE10, TE11, TE13/HET-1A	↑	β -catenin/ WISP1 signaling pathway	Cell proliferation, Invasion, Poor prognosis.	[26]
	Tissue: 93 ESCC and ANTT Cell lines: Eca109, TE-1, EC-1, ESC410/*HET-1A*	↑	↓miR-200b ↑IGF2BP2/TK1 Axis	Migration, Invasion, Tumorigenesis	[60]
Gastric cancer	Tissue: 85 GC and ANTT	↑	-	Lymph node metastasis Distant metastasis, Poor prognosis	[61]
	Tissue: 108 GC and ANTT Cell lines: Tu: SGC7901, MKN45, BGC-823, MKN-28/N: GES-1	↑	↑*ZEB2*, ↑*VIM*, ↑*CHD1*, ↑*CHD2* ↑*EZH2*, ↑3K27me3 ↑LSD1, ↑LATS	Poor prognosis Proliferation Migration, Invasion, EMT	[62]
	Tissue: 60 GC and ANTT Cell lines: GES-1, RGM-1, SGC-7901, SNU-1, HGC-27	↑	↑mTOR signaling	Proliferation, Metastasis	[63]
	Serum: 167 GC and 110 controls	↑	-	Tumor stage, Invasion, Lymph node metastasis	[64]
Hepatocellular carcinoma	Tissue: 50 HCC and ANTT Cell lines: Tu: HepG2, HEP3B, HCCLM3, HuH7/L02	↑	-	Proliferation, Migration Antiapoptotic	[65]
	Tissue: 60 HCC and ANTT Cell lines: SMMC-7721, PLC/PRF/5, Huh7, SK-Hep-1, Hep3B	↑	↑FOXM1 ↓ miR-34a	Poor prognosis, Proliferation, Tumor growth, Antiapoptotic	[66]
	Tissue: 96 HCC and ANTT Cell lines: HepG2, SMMC772, MHCC97H /MIHA	↑	↑CDH1, ↑CDH2, ↑VIM, ↑SNAI2	Poor prognosis, TNM stage, Vascular invasion, Alcoholism history, Migration, Invasion, EMT	[67]
	Cell lines: SMMC7721, SK-hep1, HepG2, Huh7/L02	↑	↑NDRG1 promoter	Proliferation, Migration, Invasion	[68]
	Cell lines: Hep3B, HepG2, and THLE-3, MHCC97H	↑	↓miR-145 ↑*MDM2*	Proliferation, Metastasis	[45]
	Tissue: 61 HCC and ANTT Cell lines: HepG2 HCCLM3	↑	↓miR- 4496 ↑ELAVL1	Advanced stage, Venous invasion, Migration, Invasion	[69]
Pancreatic cancer	Tissue: 80 PDAC and ANTT Cell lines: PANC-1, SW1990, PC-3/HPDE6-C7	↑	↑KRAS, ↑MEK/ERK	Poor prognosis, Proliferation, Invasion Tumor growth	[70]
Glioma	Cell lines: A172, U87-MG, U251, T98G/HUVECs		↑VEGF, ↑TGFβ, ↑FGF	Angiogenesis, Migration Proliferation	[71]
	Tissue:134 Glioma and ANTT Cell lines: U87-MG, U251	↑	↑Wnt/β-catenin	TNM stage, Proliferation Cell cycle, Migration Tumor growth	[72]
	Tissue: 134 Glial tumors and ANTT Cell lines: U87, U251, A172, SHG44/Normal human astrocyte cell line	↑	↑CDH1, ↑CDH2, ↑VIM, ↑TWIST, ↑ SNAI1	Poor prognosis, Tumor grade, Tumor size, Proliferation, Migration, Invasion, Apoptosis, EMT	[73]
	Tissue: 74 PA and ANTT Cell line: HP75	↑	↑E2F1 ↑PTTG1	Poor prognosis, Proliferation, Antiapoptotic, Cell cycle, Migration, Invasion	[74]
	Tissue: 138 Gliomas and ANTT Cell lines: U251, U87, A172, SHG44.	↑	↓ miR-424 ↑CHK1	Proliferation, Invasion, Migration via miR-424 sponging and CHK1 regulation	[44]
	Cell lines: A172, U251	↑	↓ miR-424 ↑ VEGFA	Proliferation, Migration, Angiogenesis	[43]
Neuroblastoma	Tissue: 96 Neuroblastomas and ANTT Cell lines: SH-SY5Y, SK-N-SH/HUVEC	↑	↓P53 ↑BCL-2	Antiapoptotic, Cell growth, Poor prognosis	[75]
Lung cancer	Tissue: 57 NSCLC and ANTT cell lines: A549, NCI-H1975, NCI-H358, NCI-H1650, NCI-H1299, SK-MES-1, Pc-9/HBE	↑ ↑ H1975, Pc9, NCI-H358 ↓NCI-H1299 NCI-H1650, A549 SK-MES-1		Proliferation, Invasion	[76]
	Tissue: 112 SCLC and ANTT Cell lines: DMS-53, H446/16 HBE	↑		Poor progression, Clinical stage, Tumor size, Distant metastasis Proliferation, Invasion	[77]
	Tissue: 36 NSCLC and ANTT Cell lines: NCI-H1975	↑	↑Wnt/β-catenin	Tumor size, Lymph node metastasis	[78]
	Cell lines:A549, SPC-A- 1, H1395, H441, H1975/BEAS-2B	↑	↑FOXC1 ↓ miR-23b-5p	Proliferation, Migration	[42]
	Tissue: 32 NSCLC and ANTT	N/A	-	-	[79]
	Serum: 438 LC and 438 controls	↑	-	-	[80]
Osteosarcoma	Tissue: 50 OS and ANTT Cell lines: SAOS-2, MG63, U2-OS/Normal osteoblast cell line	↑	↑GSK3β/β-catenin	Tumor size, Poor prognosis, Proliferation	[81]
	Tissue: 40 OS/ANTT Cell lines: SOSP-9607, MG-63, U2OS, SAOS-2/hFOB	↑	↑LATS2, ↑*MYC* ↑CDH1, ↑CHD2, ↑SNAI1	Poor prognosis, Proliferation. EMT	[82]
	Cell lines: SOSP-9607, MG-63, U2OS, SAOS-2 /hFOB	↑	↓miR-143, ↑FOSL2	Proliferation, Metastasis	[83]
Thyroid cancer	Tissues: 30 pairs TC and ANTT (papillary, follicular, and anaplastic) Cell lines: TPC- 1, TH83, IHH4, FTC- 133, FTC- 238/Nthy-ori3-1	↑	↑Wnt/β-catenin	Proliferation, Migration, Invasion, Apoptosis	[84]
	Tissue: 60 anaplastic and papillary TC and ANTT Cell lines: TC cell lines	↑	-	Doxorubicin and cisplatin resistance, Increased tumor size, Poor prognosis,	[85]
Multiple myeloma	Serum: 106 MM and 106 matched normal controls	↑	-	ISS stages, Renal dysfunction, Serum creatinine	[86]
Acute myeloid leukemia	Bone marrow samples: 46 patients and 46 healthy volunteers Cell lines: KG-1	↑	Cell cycle arrest in S phase	Cellular proliferation, Poor prognosis	[87]
Breast cancer	Tissue: 997 BC and ANTT and 56 BC and ANTT Cell lines: MDA-MB-231, MDA-MB-436	↑	-	Poor prognosis, Therapeutic response	[88]
	Tissue: 67 BC and ANTT Cell lines: MDA-MB-231, MCF-7/Hs578Bst	↑	↑Wnt/β-catenin ↑*CCND1* ↑MYC	Poor prognosis, Proliferation, Invasion, Tumorigenesis	[30]
	Cell lines: MCF-7, T47 D tamoxifen resistant/MCF-7, T47D–tamoxifen responsive	↑ tamoxifen-resistant cell lines		Suppressing CCAT2 expression improves sensitivity to tamoxifen in resistant cells	[89]
	Tissue: 48 BC and ANTT	↑	-	Lymph node metastasis	[90]
	Tissue: 67 BC and ANTT Cell lines: MDA-MB-231, MCF-7/MCF10A	↑	↑P15 ↑EZH2	Poor prognosis, Proliferation, Invasion, Cell cycle, Tumor growth	[91]
	Tissue: 60 BC and ANTT Cell lines: LCC9, MDA-MB-231 MCF-7/HCC1937	↑	↑TGF-*β*, ↑Smad2, ↑α-SMA	Lymph node metastasis Proliferation, Invasion, Migration, Apoptosis, Cell cycle	[92]
Endometrial cancer	Tissue: 30 EC and ANTT Cell lines: HEC-1-A and RL95-2	↑	↓miR-216b ↑PI3K/AKT	Proliferation, Migration, Invasion, Apoptosis	[41]
Cervical cancer	Tissue: 123 SCCC and ANTT	↑	-	FIGO stage, Lymph node metastasis, Cervical invasion, Poor prognosis	[93]
	Cell lines: CaSki, HeLa, SiHa	↑	-	Proliferation, Apoptosis	[94]
	Serum: 115 SCCC, 79 CIN, and 110 healthy controls	↑	CCAT2, LINC01133, LINC00511 upregulated in serum of SCCC and CIN patients.		[95]
	Tissue: 30 SCCC and ANTT Cell lines: GH329, CaSki, HeLa, SiHa, C4-1. Xenografts: 2 groups pSilencer, pSilencer/sh-CCAT2	↑	↓miR-493-5p ↑CREB1	EMT, Proliferation	[96]
Ovarian Cancer	Tissue: 31 EOC and ANTT Cell lines: SKOV3, MC685, A2780, HO8910/IOSE 386	↑	↓miR-424	Proliferation, Apoptosis	[39]
	Cell lines: SKOV3, A2780, HO8910/HOSE, HUM-CELL-0088	↑	↑CDH1, ↑CHD2, ↑SNAI1, ↑SNAI2, ↑TWIST1, ↑Wnt/β-catenin	Migration, Invasion, EMT	[28]
	Tissue: 109 EOC and ANTT Cell lines: SKOV3, IGROV1, A2780, OVCAR3/HOSE 6.3	↑	-	FIGO stage, Tumor grade, Distant metastasis, Poor prognosis, Proliferation Migration, Invasion	[97]
	Cell lines: SKOV3 and A2780	↑	↑TCF7L2, ↑MYC	Vitamin D suppresses CCAT2 expression	[98]
Prostate cancer	Tissue: 96 PC and ANTT Cell lines: DU-145, 22RV1/WPMY-1	↑	-	Poor prognosis, Proliferation, Migration Invasion, EMT	[99]
Renal cell cancer	Tissue: 61ccRCC and ANTT Cell lines: 786-O, AHCN ccRCC/HK-2	↑	↑Wnt/β-catenin	Poor prognosis, Proliferation, Migration, Apoptosis, Invasion	[29]
Bladder cancer	Tissue: 48 BC and ANTT Cell lines: SV-HUC-1/T24, 5637	↑	-	Tumor grade, TNM stage, Proliferation, Migration, Apoptosis	[100]

AML, acute myeloid leukemia; ANTT, adjacent non-tumor tissue; BC, bladder cancer; CC, cervical cancer; ccRCC-clear cell renal cell carcinoma; CRC, colorectal cancer; EC, endometrial cancer; HCC- hepatocellular carcinoma; MS status, microsatellite status; EMT, epithelial–mesenchymal transition; GC, gastric cancer; EOC, epithelial ovarian cancer; ESCC, esophageal squamous cellular carcinoma; MM, multiple myeloma; PC, prostate cancer; PDAC, pancreatic ductal adenocarcinoma; TC, thyroid carcinoma.

**Table 2 ijms-22-12491-t002:** CCAT2 regulatory roles on therapeutic resistance in human cancers.

Cancer Type	Role of CCAT2 in Therapeutic Resistance	Ref.
Thyroid cancer	Upregulation is associated with chemoresistance to doxorubicin and cisplatin.	[85]
Colorectal cancer	Upregulation is associated with chromosomal instability and chemotherapy resistance to 5-fluorouracil and oxaliplatin.	[32]
Lung cancer	Presence of the rs6983267 SNP was associated with reduced hematological toxicity to platinum-based chemotherapy and platinum-based chemotherapy response.	[123,125]
Breast cancer	Upregulation enhances tamoxifen resistance in breast cancer cell lines.	[89]
Glioblastoma	Upregulation in glioblastoma cell lines increases resistance to teniposide, temozolomide, vincristine, and cisplatin.	[44]
Esophageal squamous cell carcinoma	Upregulation promotes radiotherapy resistance in ESCC cell lines by inhibiting miR-145, the expression level of P70 ribosomal protein S6 kinase 1, p53, and p21.	[59]

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
