# Peer review of "The Roles of the Colon Cancer Associated Transcript 2 (CCAT2) Long Non-Coding RNA in Cancer: A Comprehensive Characterization of the Tumorigenic and Molecular Functions"

_ijms, 2021, doi:10.3390/ijms222212491_

Round 1
Reviewer 1 Report
ijms-1463787
General comments
The author describes molecular significance of a long non-coding RNA, CCAT2 in various carcinogenesis. Although this manuscript contains well written text, high-quality figures and comprehensive tables, there are some flaws regarding the style of writing.
Specific Points
- Throughout the manuscript, the style of writing seems to be inconsistent. Gene symbols with ALL upper cases mean human, whereas the ones with only initial upper case mean the other species.
- In the manuscript, use of Italic should be limited to specific condition, including, gene or Latin such as "in vitro." In addition, the reviewer fails to understand the difference between Wnt/β-catenin and Wnt/β-catenin.
- The relationship between MYC and c-myc remains unclear. Please clarify it.
- Line 187. Although the reviewer acknowledge the effort to reduce the redundancy, nevertheless, "MDS" should be "myelodysplastic syndrome."
- Line 223. There are double periods.
- Line 227. TGF seems to be duplicated.
Author Response
Dear Reviewer,
Thank you very much for providing your feedback, especially regarding the consistency issues for points 1 and 2. We have corrected the formatting and kept the italic naming only when referring specifically to the gene and not mRNA or the protein. Additionally, we have corrected Wnt/β-catenin name formatting, as we are referring specifically to the signaling pathway in which the proteins are involved. For point 3), we have corrected all naming variants and settled for MYC, as there is no functional difference between the two nomenclatures. The minor aspects referring to points 4-6 have been corrected. Thank you very much for pointing them out and for helping us improve our manuscript.
Reviewer 2 Report
Colon cancer-associated transcript 2 (CCAT2)was originally identified last 10 years as an oncologic lncRNA in colorectal cancer. Since this lncRNA is overexpressed in many other cancers, such as breast gastric, lung bladder cancers, hepatocellular carcinoma e.t.c. it is widely researched. The summary of these studies is included in the review presented for evaluation. It is a supplement to earlier reviews, it is interestingly written and may interest a wide group of readers from biochemists to doctors.
Minor remarks
I propose to present the data from the chapter "Therapeutic resistance" in the form of a table. This will make it easier to understand the problem and may be useful in planning a therapy.
Author Response
Dear Reviewer,
Thank you very much for your constructive feedback. We agree that summarizing CCAT2’s involvement in therapeutic resistance in the form of a table might facilitate in providing an overview of the described effects. Thus, we have followed your advice and arranged the described aspects from the text in a table, sorted by cancer type and a brief description of the process alongside the relevant references.
Thank you again for helping us improve our manuscript.